# Health Monitoring of Serial Structures Applying Piezoelectric Film Sensors and Modal Passport

**DOI:** 10.3390/s23031114

**Published:** 2023-01-18

**Authors:** Aleksey Mironov, Aleksejs Safonovs, Deniss Mironovs, Pavel Doronkin, Vitalijs Kuzmickis

**Affiliations:** 1D un D Centrs, LV-1021 Riga, Latvia; 2Institute of Materials and Structures, Riga Technical University, LV-1048 Riga, Latvia

**Keywords:** structural health monitoring, operational modal analysis, piezoelectric films, sensors, modal passport

## Abstract

Health monitoring of critical structures, that form parts of serial operating objects, is a pressing task. The Operational Modal Analysis (OMA) techniques could be the optimal solution. An inexpensive measurement system, such as the OMA, uses a lot of sensors for structural response assessment. The health monitoring of serial structures has to also consider possible deviations between samples. A solution providing the OMA application includes the compact measurement system based on piezoelectric film sensors and modal passport (MP) techniques. For validation of the proposed approach, a series of five similar composite cylinders, with a network of piezoelectric film sensors, was used. Applying modal tests on the specimens, and using OMA with MP methods, the set of typical modal parameters was determined and analyzed. The results of the study confirmed the feasibility of the sensor network and its applicability for structural health monitoring of serial samples using OMA methods. The proven effectiveness of OMA/MP techniques, combined with a sensor network, provides a prototype of intelligent sensor technology, which can be used for health monitoring of structures, including those that are part of an operating facility.

## 1. Introduction

Health monitoring of structures that are critical parts of complex transport, energy and other engineering objects is a pressing task. Wind turbines, aircrafts, ships, offshore platforms, pipelines, bridges, etc., are the most well-known examples of such structures. Most of these structures are of a standard design and are manufactured in series. To provide safety and efficiency, the technical operation of these structures is ensured through regular periodic inspections (surveys). To survey structures, non-destructive techniques (NDTs) are mainly used, like ultrasound, eddy current, X-ray and others [1,2]. To be applied on structures like wind turbines or aircrafts, these techniques require stop of operation. The use of optical NDT is increasing, including fiber optics, electronic speckle, and infrared thermography [3]; however, the use of such techniques on operating structures is limited. As condition-based maintenance, demanding permanent monitoring, is not in use, even if the inspection reveals no defects, stopping the facility and carrying out preventive maintenance is envisaged. Such an approach is costly, and, in some cases, for instance, in the case of ageing helicopters, it may take about 25 percent of the direct operating costs [4]. In between inspections the structures remain unobservable for a long time, while some latent defects may grow and increase the risk of damage. To reduce the above risks in operating structures, permanent structural health monitoring (SHM) is required, allowing reduction in maintenance and operational costs. 

Mostly, vibration-based fault detection methods are increasingly being used for operating objects [5] and for composite structures [6]. For permanent monitoring, modal analysis techniques may take prime position, as they are based on direct relations between the mechanical and modal properties of structures [7]. Any change in the structure modifies its modal parameters, expressed in frequencies, damping and shapes. Modal analysis techniques estimate modal parameters using vibrations measured by sensors and mathematical models [8]. To provide monitoring of the structure, a system based on modal analysis measures the vibrations of the structure, estimates the modal parameters and compares them with the reference values. For instance, in [9] change in the modal parameters of a welded steel frame was successfully used for damage detection. Regarding the helicopter rotor blade, the authors in [10] proposed a network of autonomous wireless sensors that allow structural monitoring. The combined sensor network was realized in [11] using both accelerometers and innovative optical Fiber Bragg sensors to study the capabilities of load monitoring and damage detection. The author in [12] used data acquisition from so called “smart layers”, based on sensor networks distributed in the structure. Successful application of the traditional modal analysis techniques has been limited regarding operating objects by the need to control the excitation.

The OMA methods use only the output signals of a vibrating structure without actuation control [13]. This is the main benefit they offer operating structures in terms of monitoring and damage detection, as the structures do not need to be stopped for inspection. The capability of OMA for modal properties identification of operating structures makes these techniques the most promising for structural monitoring and damage detection. Different OMA approaches to modal identification consider both stochastic subspace [14] and frequency domain decomposition [15] techniques. There are many cases of OMA applications [16] in aviation, for instance, for the airplane [17], for helicopter structures [18] and for blades [19], as well as for wind generator blades [20]. Successful trials attest to OMA techniques being the most promising approach to SHM of operating objects. For instance, applying OMA to signals of piezoelectric film sensors, the authors in [21,22] detected local damage of the rotating helicopter blade. Detection of local faults, and thereby allowing the implementation of predictive maintenance methods, is one of the goals of the prospective SHM system. The commercial software packages automating modal parameters estimation, for instance ARTeMIS [23], facilitate the implementation of OMA.

The discussed study stage is related to the paradigm shift in the application of OMA. The common approach of OMA application considers large and expensive structural objects, like bridges, skyscrapers or towers. To apply OMA, each new object requires a preliminary stage that includes an individual study (modal simulation and experiments), development and installation of a measurement and data development system. Then, for system operation, the researchers (modal analysts) and high-skilled specialists are required. 

A paradigm shift considers OMA application for serial structures that are parts of serial products, like transport vehicles, wind generators, aircrafts etc. The preliminary stage (research, development of the system and SW) is prepared only once and is then applied to all samples of that typical structure. This new paradigm necessitates novelties in the approach. First, a compact, cheap and user-friendly measurement system is required for wide application on serial structures. Second, the monitoring technique must take into account deviations in modal properties between particular samples of the typical structure. The traditional approach to monitoring involves controlling exceedance of permissible limits by diagnostic parameters. For an SHM system based on OMA modal parameters such as frequency, damping and modal shape play diagnostic roles. As modal parameters between serial structures in a reference state may vary, this could lead to errors in estimation of the condition applying the common permissible limits. 

This work focused on solving the basic tasks required for health monitoring of serial structures applying OMA methods:sensor type optimization for SHM application in serial structuresmethodology for considering the similarity and difference of modal properties between serial structures aiming for SHMexperimental verification of the approach to SHM that combines piezo films, OMA and the methodology of modal properties consideration for serial structuresinvestigation of the effect of differences between series samples on the variation of modal parameters and assessment of their applicability for monitoring.

## 2. Materials and Methods

### 2.1. Problem Analysis

#### 2.1.1. Compact Measurement System

The system prototypes used for trial OMA applications on large structures, like those in [24], may include hundreds of accelerometers and measurement channels that are too bulky for wide implementation. Typically, for modal testing, calibrated accelerometers are used and each one requires a dedicated channel. As the OMA technique considers an extensive sensor network, traditional sensors make the system heavy and costly. For a compact SHM system, lighter sensors and less measurement channels are desired. 

Film sensors, like strain gauges, look to be better suited to the above needs. Such sensors are much easier to fix on the structure, as they can be glued to the surface, together with wires, and covered with a protective layer. Having a low thickness of film allows them to be built into structural elements that are streamlined by flow, such as rotor blades, without significantly affecting their aerodynamic profile. The elasticity of films allows installation on curved surfaces, which are typical of many structures. There are two widely used kinds of film sensors: resistive strain gauges and piezoelectric films (piezo films). Strain gauges are widely used for stress measurements, while piezo films are applied mainly in signal and security systems. Unlike resistive gauges, piezo films do not measure static strain components and have problematic calibration. However, OMA techniques consider only the dynamic component of sensor signals and, besides, no calibration is required, as OMA uses a normalized scale. Thus, light and cheap piezo films may replace accelerometers in the measurement system for OMA purposes. There are also supercapacitors that could be integrated into fiber-reinforced polymers and used as sensors; however, this technique is not yet ready for application [25].

Application of piezo films allow simplifying of the system. A piezo film generates alternating electrical charge proportional to vibrations, while a resistive sensor requires a constant voltage supply, signal conditioning and additional electrical components like a Wheatstone bridge. This means that a piezo film does not generate a constant signal component and does not require balancing, as do bridge strain gauges. This is an important benefit for a system with hundreds of sensors. The OMA techniques do not require signal synchronization from multiple piezo films, so the limited number of channels may measure the multiple signals by applying a switch (the multi-patch OMA approach). It is important that mass production technology of piezo films provides a limited scatter of sensor sensitivity within the same production batch. The above benefits allow sensors to be fixed on composite structure, as these sensors do not “sense” the static component and do not need balancing as some strain gauges do. Glued on a surface of the structure, and covered by a protective layer, the piezo films, together with wires, form a “measurement layer”.

The piezo films allow serial switching of different sensors to the same measurement channel and this simplifies wiring. The network of piezo films may have a common bus and each sensor needs only signal wire. Varnished wires (not cables), such as those that are 0.25 mm in diameter, and piezo films (<0.2 mm thick), form the “measurement layer” that is covered by a protective layer. The latter is not thicker than the films/wires and smooths out the roughness. Trial applications of piezo films into composite blades [19,26,27] demonstrated workability of such technical solutions in operational conditions. For instance, the measurement system remained functional at rotating frequencies up to 25 rev/sec, with no discernible effect on the aerodynamic profile.

Mass application of piezo films requires improvements in their production and application technology. For example, they could be produced as ready-made clusters, each containing not only sensors, but also wires and connectors. The design, shape and other features of such clusters would take into account the specific design and operation of the object for which they are adapted. Such clusters of piezo films could be easy to mount on serial objects and ready components of the sensor network. Based on the possibilities of future piezo film production technologies, the OMA-based SHM system with piezo films may be compact and cost effective, having limited (at least two) measurement channels. 

Commercially available software for primary modal parameter estimation promotes OMA implementation. When applying such software to process piezo film signals, it is necessary to take into account their distinctions from accelerometer signals. The signal of a piezo film is proportional to surface strains or curvature, but not to acceleration. The methodical aspects of piezo film signals application for OMA, instead of vibrations, were considered earlier [26,27]. There are already practical cases of piezo film application for modal parameters estimation for a trial helicopter SHM system [21], for rotating blades [22] and as a sensor network for composite structures [28].

Taking into account the above-mentioned, a simple and inexpensive SHM, replicable for serial objects, could be created using piezo film sensors in combination with OMA estimation tools.

#### 2.1.2. Modal Variation between Serial Structures

The typical approach to SHM considers that the decision about abnormality of a particular structure may be taken if the diagnostic (modal) parameter exceeds the threshold. (Figure 1a).

The thresholds of modal parameters for particular structure are determined using simulation and tests of the structure in its reference state. In the case of serial structures, the thresholds should be common to all samples of the same design. Even a few deviations within the manufacturing process may differentiate mechanical properties of serial samples that variate its modal parameters within a deviation range (Figure 1b). Modal deviations may vary the operational life (T1, T2) for serial samples. With that in mind, the SHM system has to consider both the similarities and differences between serial samples. The universal approach for the evaluation of modal properties using the typical and individual parts of the Modal Passport (MP) was proposed in [18,25,26].

This study focused on the methodical and practical aspects of the intelligent sensing technology that applies the piezo films, in combination with OMA and MP techniques for SHM of serial samples. For experimental verification, five samples of the typical design were fabricated using the same technology and materials. Due to the manual processes the produced samples differed slightly from one another, and this was used to estimate the effectiveness of the proposed approach.

### 2.2. Application of MP Techniques

#### 2.2.1. Modal Passport

The MP approach considers the application of modal parameters for structural monitoring. The MP considers both the common modal properties of serial samples and the modal deviations between particular samples. The set of modal parameters, which are common for the design of serial samples, is the core of the MP. Aimed at SHM, the MP considers solutions of two principal tasks: accurate computation of modal parameters and their application for monitoring. For modal parameters computation, the MP considers the application of modal enhancement (see Section 2.2.3) to primary modal estimates provided by OMA. For monitoring, typical and individual parts of MP are applied.

(a)Typical passport

A typical MP is a set of modal data and procedures that are common to all samples of the typical design. The principal components of a typical passport are shown in Figure 2.

A geometrical model is built with N degrees of freedom (DOFs) matching to N sensors, which define geometrical resolution of a typical design. The typical signal data recording procedure *2* takes into account the specialty of ambient/operational excitations of the measured structural vibrations, as well as measurement and computational errors. For example, for wind turbine blades, the procedure would take into account the peculiarities of wind effects, mechanical excitations of a gearbox and a generator, etc. For a test case, the typical testing procedure, which simulates ambient excitation, complements the recording procedure. The typical OMA [19] algorithms *3* are used to estimate the modal parameters from the sensor signals. These algorithms, which are realized by some OMA estimators in Frequency and Time Domains, compute the modal parameters. The set of typical modal parameters *4* is common for all samples of the typical design. Since there are differences between the samples, each modal parameter of the typical set has a range characterized by a mean and an uncertainty. The latter depends on ambient factors that are considered by testing/recording procedures and may be used for an uncertainty check. External conditions affect the modal parameters and, in order to take them into account, the MP uses influence functions *5*. They reflect the influence of operational (equipment operation mode) and ambient (temperature, wind) factors on the modal parameters in the operating range. In the discussed study all ambient (temperature, humidity, etc.) and boundary conditions remained unchanged, so the influence functions were not considered here. The thresholds *6* characterize the boundaries of modal parameters in the definite state.

Thus, a typical MP includes a model, a test procedure, a set of modal parameters and a set of influence functions that are common to all samples of the typical design. For the monitoring task, the MP uses abnormality thresholds for each type of modal parameter of reference or other specified state.

(b)Individual passport

The individual MP reflects the application of data and the procedures of the typical MP to the particular sample (specimen). Application of the individual passport is illustrated, using the case of the particular sample testing (Figure 3).

The specimen is tested in accordance with the typical procedure 1, which, in the discussed study, describes specialties of hammer excitation and signals recording. The signals digitally recorded during a single test are the array of input data for modal estimation. Developing this data, the system processor 2 computes primary estimates of modal parameters by applying OMA algorithms to the modal model (common to all specimens). The primary estimates have a high uncertainty due to the influence of random factors, so further processing is required. From the set of primary estimates of a single test, the sustainable modes 3 are selected and grouped. The selected parameters are used for modal enhancement procedures 4 to reduce modal uncertainty. This uncertainty level is controlled by a sufficiency criterion 5, and if it is not met, the test of the specimen is repeated.

In this study, a simplified sufficiency criterion (triple test repetition) was applied that provided acceptable levels of uncertainty for the modal parameters. When the criterion was satisfied, the enhanced modal parameters were stored 6 in the modal data base. The check of monitoring rule 7 considered a comparison of the modal parameters with the typical thresholds.

#### 2.2.2. Modal Estimation

The processor of the SHM system calculates the primary estimates of the modal parameters using the signals of *N* sensors of the specimen. For higher resolution, the system uses more than one modal estimation technique (estimators). In the case of *E* estimators being used, the MP considers a calculation of *E* groups of primary estimates of the modal parameters from the data of the *i*th test. In each group, the modal parameters (frequency $f_{m}^{i}$, damping $d_{m}^{i},$ and eigenvector ${\left[s_{m}^{i}\right]}^{N}$) characterize each of the *m*th modes of the specimen. The primary estimates of modal parameters have a high uncertainty, due to the random component of the vibration signal. In tests, an uncertainty depends on manual actuation of a specimen and ambient factors, such as acoustic noises, vibrations, etc. Another cause of uncertainty is the mismatch of modal estimates that different estimators give out from the same test data. As various OMA estimators apply different identification methods, they provide a different number of modes and the modal estimates differ. For example, from the data of the *i*th test the number of modes $M_{i}^{1}$ identified by the 1st estimator may differ from $M_{i}^{2}$ of the 2nd one. Additionally, the errors of measurement and calculation affect the uncertainty of the estimates. 

Using parameters of the similar modes obtained by different estimators, and in different tests, the modal enhancement procedures reduce the uncertainty of primary modal estimates. Enhancement procedures are similar to vibration waveform enhancement in vibration diagnostics [29,30,31], but are performed in an imaginary *N*-dimensional space, rather than in the phase plane.

#### 2.2.3. Modal Enhancement

Modal enhancement, reducing the uncertainty of modal parameters, requires a sequence of eigenvectors transformations, including modal shape normalizing, modes grouping and phase alignment.

The need for *modal shape normalizing* is caused by the unequal scales of eigenvectors calculated by different estimators. To bring them to a common (−1.0… 1.0) range, each *n*^th^ element of eigenvector $s_{n}^{m}$ of the *m*th mode is normalized to the square root of the sum of squares of all *N* elements: (1)$${\bar{s}}_{n}^{m}=s_{n}^{m}/\sqrt{\sum_{n=1}^{N}s_{n}^{m}{}^{2}}$$

The *modal grouping* procedure includes the identifying similar modes, selecting the sustainable ones and clustering the similar and sustainable modes. For similarity estimation between a pair of modes, the Modal Assurance Criterion (MAC) is used [32] for primary estimated eigenvectors:(2)MAC12=|∑n=1Nsn1sn2*|2∑n=1Nsn1sn1*∑n=1Nsn2sn2*
where $s_{n1}$ and $s_{n2}$ are the compared eigenvectors, $s_{n1}{}^{*}$ and $s_{n2}{}^{*}$ are the conjugated eigenvectors.

An *MAC* value close to 1.0 indicates similar modes, while a zero value means there is no similarity. The number of required *MAC* calculations for one specimen corresponds to the squared total number of primary scores (*E*T*M*)^2^ given by *E* estimators in *T* tests. Aiming for modal enhancement in this research the modes were considered as similar, if *MAC* satisfied the condition:(3)0.98≤MAC≤0.9999

The upper limit of the interval prevented the same mode from being used for a subsequent enhancement. The lower boundary was determined by the requirement for the accuracy of modal shape description. If there were more than three similar modes, with *MAC* satisfying (3), such modes were considered as sustainable and were clustered into the groups. As a result of modal grouping, *K* unique groups of modes were selected, and each group included *M_k_* similar modes. 

Different estimators may reflect the same mode shape in opposite phases, so the *phase alignment* procedure was required. This procedure involves the check of phase compatibility between modes of a group and inversion of eigenvectors with opposite phases. To check phase compatibility, the correlation was calculated between the reference and other eigenvectors. Eigenvector values with negative correlation (close to −1.0) were considered to be opposed and were inverted. Once the eigenvector phases in the group matched, the modal enhancement could be performed.

The *enhancement* of normalized, grouped and matched modal parameters provides higher accuracy for assessments of modal parameter over $M_{k}$ modes of *k*th group.

The enhancement procedures are different for scalar eigenvalues (frequency and damping) and for eigenvectors. For *m*th mode, the enhanced mean of frequency ${\bar{f}}_{m}^{e}$ and damping ${\bar{d}}_{m}^{e}$ were calculated as the arithmetic mean of $M_{k}$ estimates of *k*th group: $${\bar{f}}_{m}^{e}=\frac{1}{M_{k}}\sum_{1}^{M_{k}}f_{m}^{k}$$
(4)$${\bar{d}}_{m}^{e}=\frac{1}{M_{k}}\sum_{k=1}^{M_{k}}d_{m}^{k}$$

Analysis of the experimental data, including [23], showed the distribution of modal estimates to be close to normal. The uncertainty of enhanced modal parameters with 99.7% probability was characterized by triple standard deviation of its $M_{k}$ estimates based on the assumption of a normal distribution: $$\delta f_{m}^{e}=3\sqrt{\frac{1}{M_{k}}\sum_{1}^{M_{k}}{\left(f_{m}^{k}-{\bar{f}}_{m}^{C}\right)}^{2}}$$
(5)$$\delta d_{m}^{e}=3\sqrt{\frac{1}{M_{k}}\sum_{1}^{M_{k}}{\left(d_{m}^{k}-{\bar{d}}_{m}^{C}\right)}^{2}}$$

The modal shape enhancement calculates the *N* elements of the enhanced eigenvector ${\bar{s}}_{n}^{m}$ by averaging $M_{k}$ matched eigenvectors of the *k*th group:(6)$${\bar{s}}_{n}^{m}=\frac{1}{M_{k}}\sum_{k=1}^{M_{k}}{\left|s_{n}^{m}\right|}_{k}$$

The uncertainty of the enhanced eigenvector is estimated for each *n*th element of enhanced eigenvector by triple standard deviation in the group of $M_{k}$ modes and is presented as a vector:(7)$${\bar{\delta s}}_{n}^{m}=3\sqrt{\frac{1}{M_{k}}\sum_{1}^{M_{k}}{\left({\bar{s}}_{n}^{m}-{\bar{s}}_{n0}^{m}\right)}^{2}}*100\%$$

As an integral estimate of the entire eigenvector of the *m*th mode, the averaged uncertainty could be estimated as:(7a)$$\delta {\bar{s}}^{m}=\frac{3}{N}\sqrt{\sum_{1}^{N}\delta {\bar{s}}_{n}^{m}{}^{2}}*100\%$$

The enhanced modal parameters computed from the test data, using Equations (4) and (6), reflected the individual modal properties of the tested particular specimen. The scores, calculated using Equations (5), (7) and (7a), characterized the level of uncertainty, with which these parameters were calculated.

#### 2.2.4. Modal Difference

The parameters estimated using the above equations could be applied for the purposes considered in this work, i.e., to assess the modal difference between serial specimens. As a reference, the modal properties of the “typical” sample were used. Such a virtual sample generalizes the properties of *Z* serial specimens. If the enhanced modal parameters ${\bar{f}}_{m}^{z}$, ${\bar{d}}_{m}^{z}$ and ${\left|{\bar{s}}_{n}^{m}\right|}^{z}$ of the *z*th specimen are computed using Equations (4) and (6), then the modal parameters of the typical sample ${\bar{f}}_{m}^{T}$, ${\bar{d}}_{m}^{T}$, ${\left|{\bar{s}}_{n}^{m}\right|}^{T}$ can be calculated: $${\bar{f}}_{m}^{T}=\frac{1}{Z}\sum_{1}^{Z}f_{m}^{z}$$
$${\bar{d}}_{m}^{T}=\frac{1}{Z}\sum_{1}^{Z}d_{m}^{z}$$
(8)$${\left|{\bar{s}}_{n}^{m}\right|}^{T}=\frac{1}{Z}\sum_{1}^{Z}{\left|s_{n}^{m}\right|}_{z}$$

Frequency and damping deviations of the *m*th mode of the *z*th specimen from the typical sample is characterized by the normalized difference: $$\Delta {\bar{f}}_{m}^{Z}={\bar{f}}_{m}^{z}-{\bar{f}}_{m}^{T}$$
(9)$$\Delta {\bar{d}}_{m}^{Z}=d_{m}^{z}-{\bar{d}}_{m}^{T}$$

The shape deviation $\Delta {\bar{s}}_{i}^{m}{}^{Z}$ of the *m*th mode of the *z*th specimen from a typical sample is calculated as the geometrical sum of elementary $\delta {\bar{s}}_{n}^{m}{}^{Z}$ deviations: (10)$$\Delta {\bar{s}}_{i}^{m}{}^{Z}=\sqrt{\sum_{n=1}^{N}\Delta {\bar{s}}_{n}^{m}{{}^{Z}}^{2}}*100\%$$

Each elementary deviation $\Delta {\bar{s}}_{n}^{m}{}^{Z}$ is the difference between the normalized *n*th elements of both eigenvectors (the compared and the typical one): (11)$$\Delta {\bar{s}}_{n}^{m}{}^{Z}={\bar{s}}_{n}^{m}{}_{z}-{\bar{s}}_{n}^{m}{}^{T}$$

As modal shape parameters have a normalized scale, the shape deviation is scaled as: $$0<\Delta {\bar{s}}_{i}^{m}{}^{Z}<1.0$$

As mechanical properties and geometry of specimens differ, modal shapes may be angularly shifted in regards to the cylinder axis. Therefore, to assess the deviation of modal shapes between the specimen and the typical sample, the modal shapes should be matched artificially. Taking the latter into account, the expressions (9)–(11) allow estimating of the deviation of modal parameters from the typical sample.

To estimate the integral modal deviation of the *z*th specimen, the deviations of all *M* modes are considered. For this purpose, the parameters of integral deviation of frequency $\Delta {\bar{f}}_{\Sigma }^{T}$, damping $\Delta {\bar{d}}_{\Sigma }^{T}$ and shape $\Delta {\bar{s}}_{i}{}^{T}$ are calculated:$$\Delta {\bar{f}}_{\Sigma }^{T}=\sqrt{\sum_{m=1}^{M}\delta {\bar{f}}_{m}^{T}{}^{2}}*100\%$$
$$\Delta {\bar{d}}_{\Sigma }^{T}=\sqrt{\sum_{m=1}^{M}\delta {\bar{d}}_{m}^{T}{}^{2}}*100\%$$
(12)$$\Delta {\bar{s}}_{\Sigma }{}^{T}=\sqrt{\sum_{m=1}^{M}\delta {\bar{s}}_{i}^{m}{{}^{T}}^{2}}*100\%$$

The above formulae allow estimation of the integral deviations of each modal parameter between specimens using the measured test data.

## 3. Experiments

The experimental phase of the study solved the following tasks:manufacturing the sample series and inspection of their structural deviationsFE modeling of the typical sample and its structural deviationsmodal tests of specimensassessment of modal properties and modal deviations of the specimensanalysis of the applicability of the MP parameters for specimen monitoring.

### 3.1. Serial Samples

#### 3.1.1. Design and Manufacturing

The cylindrical composite design (Figure 4) was chosen as the typical structure for investigation of the modal properties. 

Such a design combines features of beam and shell constructions that are widely applied in aerospace, transport and energy. The set of technical solutions, including production technology and the sensor network’s layout, was developed based on the experience of the prototype of a similar design [25]. Five samples were manufactured and specimen No.1 is presented in Figure 4b.

A key factor in applying the MP for structural monitoring is the resolution of the sensor network defined by DOF number and layout. The DOFs have to provide identification of modes correctly reflecting the waveforms of a specimen. For better identification of mostly shell modes [25], such as those of the prototype specimen, the sensors, in groups of 12 pieces, were located circumferentially in four annular sections of the cylinder. As the sensor network also included wires and connectors, the requirement of minimal influence on modal properties was met. The use of simple and inexpensive piezo films allowed high resolution, while minimizing the influence of the measuring system on the specimen’s properties. 

The typical design (Figure 4a) included a composite cylinder 1 made of fiberglass with epoxy resin, and annular flanges (top 2 and bottom 3) made of laminated plywood. The nominal height of the specimen (with flanges) was 790 mm, the outside diameter of the cylinder was 300 mm, and of the flanges it was 360 mm. The average cylinder wall thickness was 1.45 mm, and the thickness of the flanges was30 mm. The sensor network contained 48 piezoelectric film sensors 4 with connecting wires 5 and four terminals 6 for connecting the D-SUB harnesses.

Technologically, the specimen was fabricated starting from the composite cylinder, and then the network of piezoelectric sensors was mounted on its outer surface, covered by the protective composite layer, and, finally, two ring flanges were mounted. The composite cylinder was made of four layers of fiberglass with a density of 300 g/m^2^, oriented at the angles of 45°/−45° to the cylinder axis, and LG 385 epoxy resin with HG 385 hardener. Then, a network of piezoelectric sensors, with connecting copper wires, was glued with double-sided adhesive tape, as seen in Figure 4c. Four wiring harnesses of D-SUB connectors were soldered to the terminals, one of which can be seen in the photo (Figure 4c down). The sensors were evenly placed in groups of 12 around the circumference in the four circular sections of the cylinder. To protect and isolate the sensor network, a protective layer of satin weave fiberglass, with a density of 50 g/m^2^, was glued, using the same epoxy resin and a hardener. At the final stage, the cylinder was glued into the annular grooves of the flanges and the harnesses were fixed there (Figure 4b top). The nominal weight of a single completely manufactured specimen (with the flanges and connectors) was 4.37 kg.

The mass of 48 sensors, wires and accessories was 65 grams, which, in relation to the total mass, did not exceed 1.5%. Such a ratio, in combination with reliable protection of the sensor network, was considered the optimal technical solution.

#### 3.1.2. Structural Deviations between Samples

The manual manufacture of specimens led to differences between serial samples. To estimate the structural variations between specimens, an instrumental study was conducted. The analysis of measurement of the data demonstrated that the samples complied with technical requirements. At the same time, limited deviations and small differences between the specimens were identified, mainly due to hand-made technology. There were detected deviations of weight, size, misalignment, and cylinder wall thickness. 

The maximal height difference between specimens was 3 mm. The weight deviation (Figure 5a) from the reference (specimen No. 1) varied from 30 g (No. 4 & 5) to 150–180 g (No. 2, 3) which was more than 4% of the total specimen’s mass.

A misalignment of each specimen was found between the axes of the cylinder and the flanges. This misalignment, containing tilt and shift, was formed during the manual assembling of the specimens. The diagram in Figure 5b illustrates a misalignment variation (expressed in mm) between the samples. Deviation of other global dimensions could be neglected, due to their small size.

The local wall thickness variation of the cylinder part, caused by the sensor network, was of interest. For this purpose, circumferential scanning of the outer cylinder surface was conducted on the three cross sections of the cylinder part (Figure 6a). 

The distance between the base and the cylinder surface was measured, turning the specimen. The distance dependence on circle length (cylinder turnover) in the green, blue and red sections are shown in Figure 6b, which illustrates two kinds of wall thickness variation.

The smooth change (as envelope of the dependence) matches the misalignment mentioned above, while local peaks are caused by wires and sensors. In the green section (next to measurement basis), the misalignment was minimal, so, the green line in diagram (Figure 6b) mainly indicates surface roughness. The narrow “peak” (close to 600 mm circle length) was related to the trace of wiring, while the wide one (next to 400 mm) was the sensor under the protective layer. In the “red” section, misalignment played a bigger role, so the envelope part was higher than the peaks caused by wires and sensors.

The analysis of the measurement results highlighted two types of global differences between the samples that could affect the modal properties: mass (weight) and misalignment. The structural deviations related to the sensor network did not noticeably influence the modal parameters of the samples.

### 3.2. Modeling

#### 3.2.1. Typical Sample

To simulate modal properties the finite element (FE) model was created to consider the typical design of the specimens and of the massive test rig, on which the test specimen was mounted (Figure 7a). The main task of the modeling in this study was identification of frequencies and shapes of the shell modes of the cylindrical part. The shell modes and, specifically, the higher order ones were of interest as being the most sensitive to local defects that were planned to be implemented into this part of the specimens. Another task was to model the influence of global parameters on modal frequencies.

Autodesk Inventor, with built-in Autodesk Nastran-In-Cad solver, was used for modeling. This model allows modeling in a wide range of modes, including low-order ones, such as first bending in two planes. The model has 32,936 elements, including 18,505 shell ones and 85,073 nodes. The size of the elements ranged from 6 mm on the cylinder shell, to 60 mm on the U-frame of the test rig. A further decrease of elements, causing an increase in the number of nodes and elements, resulted in a significant increase in computation time without a noticeable improvement. The cylinder shell orthotropic 2D elements, with Young’s modulus of 30,900 MPa/8300 MPa, were represented as a multi-layer laminated composite element (Laminate). The Laminate consisted of 8 element layers with a thickness of 0.2 mm each, so the total thickness of the Laminate was 1.6 mm. In this model, 2 layers of Lamina with different fiber directions (+45°/−45°) simulated 1 layer of real glass fabric. The parameters of the cylinder part of the model are shown in Figure 8a. Plywood was chosen as the material for the flanges that were modeled using isotropic solid elements (Figure 8a). A specific density of 650 kg/m^3^ and a Young’s modulus of 9300 MPa were adopted for the flanges. The boundary conditions reflected the vertical cantilever attachment of the specimen (Figure 7a) by its upper flange, using 6 bolts to the horizontal beam of the U-shaped test rig. The model of the test rig, including bolts, consisted of isotropic elements with a Young’s modulus of 2.05 × 10^5^ MPa.

The frequency range of the simulation was limited by the model order that corresponded to the sensor layout. The circular arrangement of 12 sensors, reflected by 12 DOFs of the model (Figure 4a), allowed identification of the maximum 6th order mode in the circumferential direction. Along the height of the cylinder, four DOF (4 sensor rows) allowed identification of the 2nd order mode. For the cylindrical part of the sample, modes were classified according to the number of half-waves in the longitudinal (*n*) direction and of waves in the circumferential (*m*) direction and denoted by (*n;m*). Thus, the lower order mode was identified by (1;1) and the higher order mode by (2;6).

Regarding the FE modal model optimization, the frequency of the highest predicted mode (2;6) varied in the range of 320–400 Hz, so the upper boundary of the frequency range was limited to 400 Hz. The initial modeling stage provided more than 30 modes in the above range, including combined test rig and specimen modes. Due to the symmetric structure of the specimen, practically all computed modes were coupled. The model optimization was performed using comparative analysis of simulated and experimentally estimated modes with modal order from 1;1 to 2;6. In this range, there were 18 modes identified, the simulated and experimentally measured frequencies of which were used for optimizing the model. The optimization criterion was calculated as the sum of frequency errors (between measured and simulated) for all modes. The model with the lowest sum of frequency errors was chosen as the optimal one.

The numbers, designations and frequencies of the simulated modes are given in Table 1, columns 1–3. Two bending modes of the model, with vibrations perpendicular to the rig plane (1;1) and along the plane (1’;1), had the lowest order. The remaining modes had shell shapes, examples of which are illustrated in Figure 8c.

#### 3.2.2. Modeling of Deviations

The second stage of modeling was carried out to assess the influence of structural differences between the specimens on the modal parameters. Each FE model was calculated for the largest deviation of one of the varying parameters (mass, height or misalignment) found. An increase in material density was used to assess the effect of a 4% mass deviation. Two ultimate versions of density increase were considered, either only in the cylindrical part or only in the flanges. 

The axis shift (1.57 mm) and the tilt (1.65 mm) were modeled to assess the effect of misalignment. The impact of height change was achieved by increasing the height of cylindrical part by 3 mm. The effect of each type of deviation on modal frequency is presented in columns 4 to 7 of Table 1. As can be seen, mass deviation had the greatest impact. If mass grew due to uniform change of cylinder material density (column 4), the shell modes changed almost equally and only bending modes changed less. If the mass changed due to flanges density (column 5), the bending modes mainly changed, while the shell modes remained almost unchanged. Changes in the sample height (column 6) had less effect on all modes, and misalignment (column 7) even less effect.

The cumulative variation of modal frequencies (up to 2%) were caused mainly by the mass and structural deviations of the specimens. These simulations indicated potential modal frequency variation between particular samples of the same construction.

### 3.3. Modal Testing

#### 3.3.1. Test Rig and Measurement System

For modal testing of the specimens, the test rig (Figure 7a) and the data measurement system were used. 

The U-shaped test rig consisted of thick-walled rectangular tubes assembled with fasteners. The test rig was mounted on a vibration-isolated base. The weight of the rig was 156 kg, which provided a 35:1 ratio to the specimen weight. A square plywood flange, fixed with four screw ties, was placed on the underside of the upper crossbar. The specimen was attached to the square flange cantilevered by its upper flange using 6 bolts. 

The measurement and data recording system included the following:the sensor network of a specimen with 48 piezoelectric film sensors (type DT1-028K N/TH), the signals of which were routed to 4 D-sub connectors;4 cables connecting the sensor network to the measurement unit;a 48-channel measuring unit, 3660-C, with 4 modules, 3053-B120 (Brüel and Kjaer);a laptop computer with software for vibration recording.

#### 3.3.2. Testing, Measurements and Primary Estimates

The methodology for modal testing of the specimen considered the procedure of excitation, the data recording procedure and computation of the primary estimates of modal parameters.

Exciting blows on the sample, at intervals of at least 2 seconds, were applied manually with a plastic hammer. The operator hit the bottom flange (item 1 in Figure 7b) alternately in the radial and vertical directions with a gradual relocation of the actuation point around the flange. The impact force range and time intervals between impacts were determined at a preliminary test stage, varying the amplitudes and damping durations of the signals. The duration of the single test (repetitive excitation and registration procedure) of a specimen was 120 s.

The signals of the sensor network were transmitted via cables (item 2, Figure 7b) to the input of the 48-channel measuring unit (item 3). The unit provided conditioning and sampling of signals with a frequency of 3200 Hz. For each specimen, the single test procedure was repeated three times and the recorded sensor signals were digitally stored as files, which were the input data for modal parameters estimation.

The primary modal estimates were calculated with ARTeMIS software using test data. ARTeMIS provides an implementation of the most usable OMA techniques, including Enhanced Frequency Domain Decomposition, Canonical Variate Analysis and Unweighted Principal Component analysis. For modal shapes illustration, the program uses the geometric model (Figure 7c) of a specimen. The DOF vectors of the model are oriented normally to the surface and reflect the magnitude and direction of normalized curvature with the appropriate sign, providing a spatial reference of the eigenvector to the sensor signals.

For each identified mode, ARTeMIS computed primary estimates of frequency, damping and shape. The frequency and damping provided an indication of standard deviation. The mode shape was represented by an eigenvector reflecting magnitudes and directions of 48 DOF at the sensor locations.

The MP methodology (Section 2) for further modal development used the above primary estimates calculated by all estimators in different tests. The MAC (Equation (1)), using modal shapes, provided identification of primary modal estimates. Other methods of identification, e.g., by modal frequencies, were difficult, due to proximity of neighboring modes and scatter of frequencies between the estimators and repeated tests. The number of primary estimates provided by five OMA estimators of ARTeMIS for three tests varied, 300 ± 20, so the total number of calculated MAC for one specimen varied in the range (78… 102) × 10^3^. For similar modes, where the MAC satisfied condition (3), modal enhancement procedures were applied according to Formulas (4)–(7).

### 3.4. Modal Estimation of Specimens

Using the primary estimates of specimen tests the enhanced modal parameters were computed. Eigenvalues of modal frequencies (Table 2) and damping (Table 3) are presented in columns 4–8. Some paired even-order modes (1;6’, 2;4’ and 2;6’) appeared weakly during the tests, so they were not considered in the following analysis. Finally, 15 sustainable modes were identified from the experimental data of 5 specimens (except for mode 1 in specimen 4) that were considered to be the set of typical modes. Column 9 of Table 2 and Table 3 shows the mean (frequency or damping) of modal parameters averaged over the 5 specimens. The uncertainty of these averaged modal property computations is shown in column 11 of both tables. Column 10 shows the standard deviation between the modal parameters of the specimens. 

Note that the values in columns 9 and 11 no longer characterized a particular sample, but rather the modal properties common to all specimens. Such a set of generalized modal parameters, obtained from experimental evaluations of a sample series, could be considered a modal model of the “typical” sample. The typical sample, calculated from experimental data, represented the modal properties common to the five specimens and was a part of a typical MP. For each mode, column 9 includes the typical mean, and column 11 characterizes an uncertainty of the modal parameter. For instance, for the mode *(1;6)* the “typical” frequency was 327.0 Hz, with uncertainty of about 1.0 Hz (Table 2). Comparative analysis between the experimentally computed and the modeled modal shapes confirmed compatibility; however, its frequencies differed. In order to assess the error between simulated and averaged (typical) frequencies, column 12 of Table 2 presents the normalized (%) differences calculated by Formula (9). 

Deviations of modal shapes between specimens were estimated based on comparison to appropriate typical (averaged) shapes. The parameter of modal shape variation $\sigma {\bar{s}}_{i}^{m}{}^{Z}$ estimates the normalized shape deviation of *m*th mode of *z*th specimen using Formulas (10) and (11), and taking into account the deviations of each DOF of the eigenvector. 

Table 4 contains the estimated deviation of each mode shape for every specimen and the averaged uncertainty of such estimates.

The integral modal parameters were more convenient for assessment of modal deviations between specimens. The latter considered the summed parameter deviations of all sustainable modes that were calculated according to Formula (12). A diagram of the integral parameter variation between specimens, in respect to a typical one, is shown in Figure 9. The correlation between experimentally obtained integral modal parameters, on one hand, and mass and dimensions of the samples (studied in Section 3.1.2), on the other hand, was calculated. Thus, the correlation of modal damping (red bars on Figure 9) with mass of specimens (Figure 5a) was as high as 0.85, while the correlation with misalignment (Figure 5b) was 0.33. The parameter of modal shape variation (grey on Figure 9) correlated well (0.87) with specimen misalignment, but not with specimen mass (0.37). At the same time, the integral frequency parameter had no meaningful relation with the mass or misalignment of specimens.

## 4. Discussion

The sensor network (48 DOFs) of a specimen, in combination with OMA and MP techniques, allowed reliable two-dimensional modal identification with the highest 2;6 order. The sensing systems of all five specimens worked smoothly while testing, and allowed a comparative analysis between each specimen’s modal properties. 

To simulate structural differences between specimens, the initial FE model was optimized using the experimentally obtained modal parameters. The serial (averaged) modal frequencies of the five specimens were used for optimization of the model. The sum of errors between 15 identified simulated and averaged modal frequencies were used as the criterion of optimization. The error (column 12 of Table 2) of higher order modes (1;3 and higher) did not exceed 5%, while lower order modes had higher errors, especially the 1st order (bending) modes, reaching 12…14%. Since the errors for higher modes, which were of interest in this study, were satisfactory, no further optimization of the model was carried out. A high similarity between the simulated and experimentally obtained modal shapes also confirmed the correctness of the optimized model, especially for shell modes. From a practical point of view, it was concluded that this model was satisfactory, because it predicted all sustainable modes revealed in the tests. The optimized model was used for simulating the mass-dimensional differences between specimens. Simulated change of mass (4%) affected the frequencies of bending or shell modes by about 1% (Table 1), depending on where it was concentrated. Height differences between specimens had less effect (0.03…0.65%) and misalignment even less (0.01…0.39%) effect. 

The comparison of experimentally obtained modal parameters between serial specimens and simulated ones were of interest. Frequencies of the similar modes varied between specimens (column 10 of Table 2) with a standard deviation of 0.3…14.7 Hz. The higher the modal order was (1;5, 2;6, 1;6), the higher was the frequency deviation. The maximal deviation normalized to the “typical” frequency mean reached 4.5%. In comparison to simulated deviations of frequency, the experimentally determined ones were higher, which meant the presence of factors not taken into account in the modeling. The damping parameter varied between specimens (column 10 of Table 3), and even more so in percentage. In comparison to frequency and damping, the normalized parameter of modal shape deviation between specimens had the smallest scale (Table 4) and its standard deviation did not exceed 1.2%. There were two tendencies that should be noted for modal shape deviations. First, as the mode order became higher, the shape deviation between specimens (from “typical” one) was smaller. This meant that the higher mode shapes were less sensitive to a specimen’s global parameters variation. Second, the standard deviation and uncertainty of modal shape parameter estimation were independent of modal order. The exception for higher frequency modes (2;4 and 2;6) was probably related with the limitation of the applied modal model (limited DOF number) of the sensor network of a specimen. Small sensitivity of modal shape parameter to global parameters of specimens demonstrated effectiveness of a parameter to local damage identification precisely for serial structures. 

So, for monitoring serial thin-walled structures, the parameter of modal shape deviation was the most applicable. 

The integral modal parameters (Figure 9) allowed an overall assessment of modal property deviations between specimens (in relation to a typical sample). These parameters clearly illustrated the total scale of modal parameter deviations between specimens: namely, frequency (5–14%), damping (67–102%) and shape (2–4%). The standard deviations of modal parameters (columns 10 of Table 2, Table 3 and Table 4) allowed the determining of the lower limit of their sensitivity to damages in series samples. At a 95% probability level, the modal change sufficient to detect damage had to exceed double standard deviation of the parameter. For example, in order to ensure the detection of damage by modal frequency parameter in any serial sample, it was necessary that the defect caused a frequency change in mode 1;3 of at least 2.8%, and in mode 1;6 of more than 29.4% (see Table 2). Such large-scale changes of modal frequency would have to correspond to very serious structural changes. Even higher (in percentage terms) was the lower limit of sensitivity of the modal damping parameter (Table 3). In contrast to global frequency and damping parameters, the difference in shape between similar modes was much smaller. Therefore, for damage detection (at 95% probability) the shape change of any mode must not be less than 1.3%, and, for some modes, 0.3–0.5% was enough (Table 4). Thus, modal shape parameters had the lowest sensitivity threshold for damage detection in a serial sample and, accordingly, had the greatest diagnostic capabilities.

High correlation between modal parameters (damping, shape) and global structural features (mass, misalignment) of specimens illustrated the diagnostic capabilities of modal parameters. On the other hand, the lack of such correlation for the integral frequency parameter indicated low sensitivity of modal frequencies to the structural deviation of the specimens. The results of comparison confirmed that integral modal parameters could also be used for monitoring purposes, especially for the identification of global structural changes. 

## 5. Conclusions

The study considered a new paradigm for the application of Operational Modal Analysis (OMA) methods in structural monitoring and damage detection of serial structures. Structural differences between serial samples and related modal parameter deviations were the specific focus of the research. The proposed technical and methodical solutions were oriented to provide a potentially compact and cost-effective system that could be applied to serial structures.

The piezo electric film sensors, as the core of technical solutions, demonstrated capability to form a complete sensor network and to provide signals sufficient for OMA. It was noted that, for mass application of piezo films, new technologies need to be developed, like the production of ready-made segments (clusters) for sensor networking on objects. 

As the methodological basis of the study of the modal properties of serial structures the already known OMA algorithms of modal estimation, and the modal enhancement procedures of the proposed modal passport (MP), were applied. For experimental research on the capabilities of the piezo films and the methods, a series (five) of similar composite specimens were manufactured. The experimental data obtained from tests on the specimens facilitated comparison of the diagnostic capabilities of modal parameters, i.e., frequency, damping and shape. The modal shape parameters had the lowest sensitivity threshold for damage detection in serial samples and, accordingly, had the greatest diagnostic capabilities.

For global changes identification, the integral parameters, combining the properties deviations of all identified modes, could be used. 

The proven effectiveness of OMA and MP techniques, combined with a piezo film sensor network, may be a prototype for intelligent sensor technology and could enable it to be used for monitoring structures that are part of an operating facility.

## Figures and Tables

**Figure 1 sensors-23-01114-f001:**
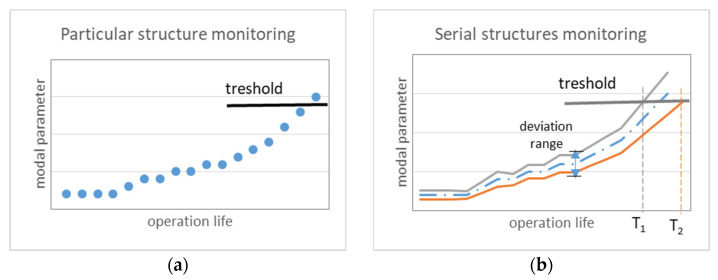
Parameter elevation for health monitoring: (**a**) particular structure, (**b**) serial structures (with deviation range).

**Figure 2 sensors-23-01114-f002:**
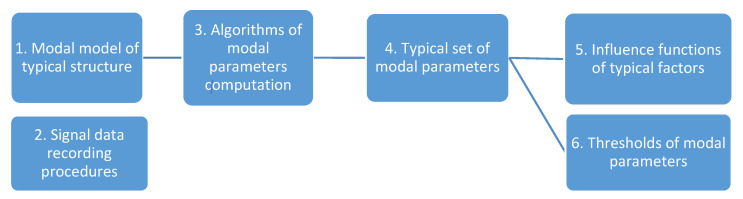
Typical modal passport.

**Figure 3 sensors-23-01114-f003:**
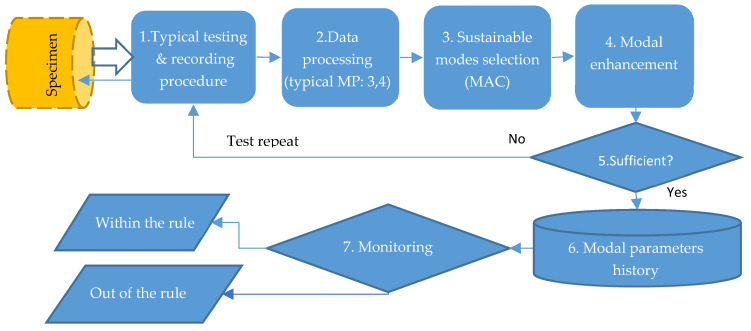
Procedures of individual MP.

**Figure 4 sensors-23-01114-f004:**
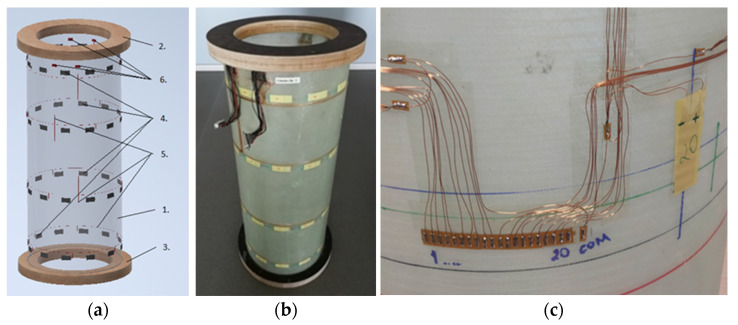
Specimen design (**a**) photo of the specimen No. 1 (**b**) sensor network preparation (**c**).

**Figure 5 sensors-23-01114-f005:**
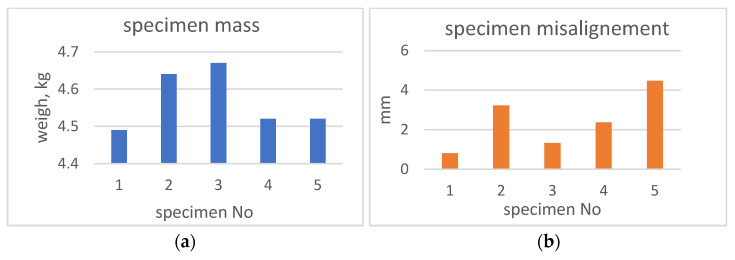
Illustration of samples deviations: weight (**a**), misalignment (**b**).

**Figure 6 sensors-23-01114-f006:**
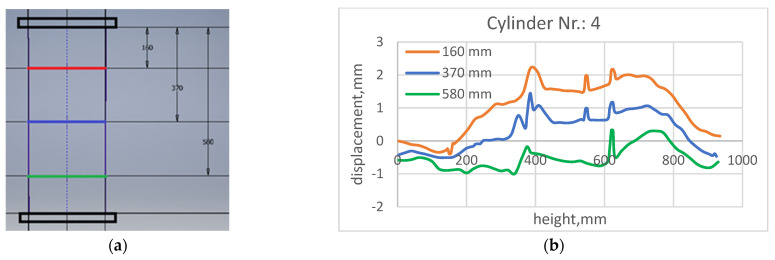
Variation of specimen wall thickness: measurement sections (**a**) circumferential scans (**b**).

**Figure 7 sensors-23-01114-f007:**
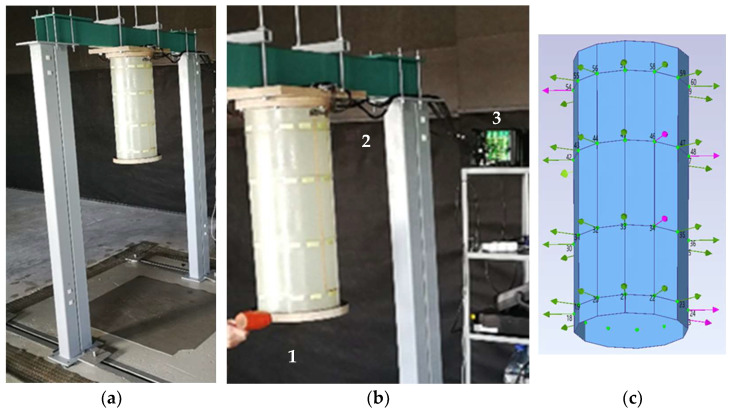
Tested specimen: test rig with specimen (**a**), actuation and measurement (**b**) geometrical model of the specimen (**c**).

**Figure 8 sensors-23-01114-f008:**
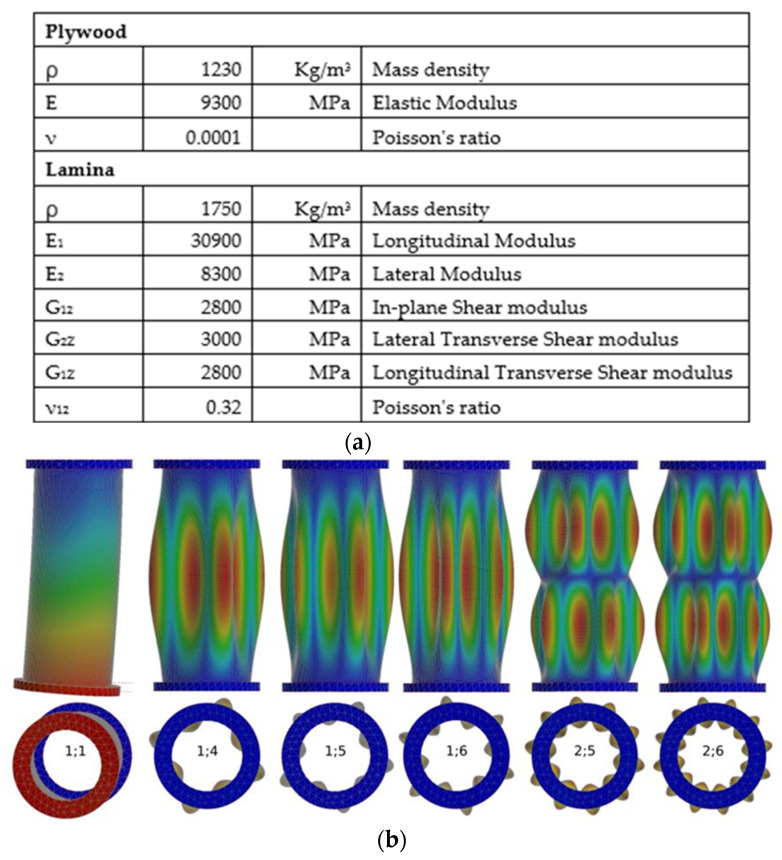
Parameters of the model (**a**), samples of simulated bending and shell modes (**b**).

**Figure 9 sensors-23-01114-f009:**
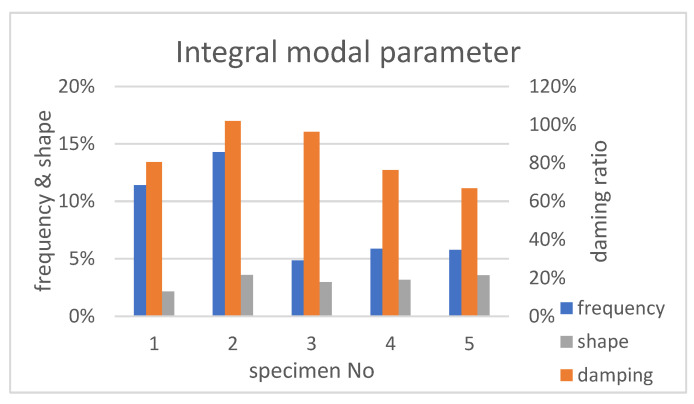
Integral modal parameter variations between specimens.

**Table 1 sensors-23-01114-t001:** Simulated frequencies of typical sample and influence of deviations.

Mode		FE Model Frequency, Hz	Frequency Change, %			
No	Type		Mass Change		Height Change	Misalign Ment
			Cylinder	Flanges		
1	2	3	4	5	6	7
1	1;1	89.59	−0.33%	−1.02%	−0.46%	0.12%
2	1;1’	122.16	−0.34%	−1.19%	−0.59%	0.06%
3	1;4	175.98	−0.98%	0.00%	−0.38%	0.07%
4	1;4’	176.06	−0.98%	0.00%	−0.38%	0.05%
5	1;3	194.222	−0.98%	0.00%	−0.65%	0.39%
6	1;3’	194.415	−0.98%	0.00%	−0.65%	0.27%
7	1;5	223.262	−0.98%	0.00%	−0.13%	−0.02%
8	1;5’	223.275	−0.98%	0.00%	−0.13%	−0.04%
9	1;2	294.136	−0.90%	−0.05%	−0.06%	0.30%
10	1;2’	295.908	−0.96%	−0.03%	−0.06%	0.38%
11	1;6	308.002	−0.99%	−0.63%	−0.45%	0.01%
12	1;6’	308.004	−0.99%	−0.68%	−0.45%	0.00%
13	2;5	312.95	−0.99%	−0.05%	−0.30%	0.00%
14	2;5’	313.016	−0.99%	−0.05%	−0.58%	−0.01%
15	2;4	345.91	−0.98%	−0.42%	−0.25%	0.18%
16	2;4’	346.158	−0.98%	−0.42%	−0.25%	0.14%
17	2;6	357.788	−0.34%	−0.20%	−0.03%	0.01%
18	2;6’	357.797	−0.98%	−0.20%	−0.52%	0.00%

**Table 2 sensors-23-01114-t002:** Modal frequency.

FEM Model			Frequency, Hz (Experimental Estimates)								Error
Nr	Order	F, Hz	Sp 1	Sp 2	Sp 3	Sp 4	Sp 5	Typ. Sp	STD (1–5)	Unc-ty	
1	2	3	4	5	6	7	8	9	10	11	12
1	1;1	89.62	78.3	78.1	78.8	-	78.4	78.4	0.3	2.1	−13%
2	1;1’	122.52	108.4	105.9	104.0	104.4	104.8	105.5	1.8	0.9	−14%
3	1;4	177.02	175.4	185.5	179.5	175.2	174.5	178.0	4.6	0.5	1%
4	1;4’	177.10	177.6	185.5	178.9	177.2	175.9	179.0	3.8	0.3	1%
5	1;3	195.11	187.1	185.7	183.3	185.3	185.0	185.3	1.4	1.8	−5%
6	1;3’	195.31	190.5	188.4	188.4	181.0	185.0	186.7	3.7	2.9	−4%
7	1;5	224.82	224.7	248.4	238.7	239.7	230.2	236.3	9.2	1.1	5%
8	1;5’	224.83	226.3	249.2	239.2	230.8	232.4	235.6	8.9	0.7	5%
9	1;2	295.43	267.5	260.6	257.0	262.8	261.9	262.0	3.8	0.8	−11%
10	1;2’	297.31	284.0	267.7	263.5	267.2	266.2	269.7	8.2	2.7	−9%
11	1;6	310.23	311.4	349.2	333.7	319.9	321.0	327.0	14.7	1.0	5%
12	2;5	314.99	317.9	332.4	324.2	316.9	315.6	321.4	7.0	1.0	2%
13	2;5’	315.06	323.4	334.2	324.2	316.9	319.2	323.6	6.7	2.8	3%
14	2;4	347.96	355.1	356.2	348.6	347.5	344.5	350.4	5.0	2.4	1%
15	2;6	360.37	358.5	397.1	377.3	368.2	366.6	373.5	14.7	2.3	4%

**Table 3 sensors-23-01114-t003:** Modal damping.

Mode		Damping, % (Experimental Estimates)							
Nr	Type	Sp 1	Sp 2	Sp 3	Sp 4	Sp 5	Typ.Sp	STD (1…5)	Unc-ty
1	2	4	5	6	7	8	9	10	11
1	1;1	1.1	0.7	5.3	-	4.3	2.8	2.28	2.72
2	1;1’	1.7	2.0	1.5	1.6	1.7	1.7	0.11	2.00
3	1;4	0.7	0.7	1.0	0.7	0.7	0.8	0.18	0.14
4	1;4’	0.8	0.7	0.8	0.7	0.8	0.8	0.05	0.11
5	1;3	0.7	0.7	0.8	0.9	0.9	0.8	0.12	0.53
6	1;3’	1.1	0.8	0.8	1.1	0.9	0.9	0.16	0.57
7	1;5	0.7	0.8	0.7	0.6	0.7	0.7	0.09	0.61
8	1;5’	0.6	0.7	0.7	1.0	0.6	0.7	0.24	0.62
9	1;2	1.2	1.0	1.2	1.2	1.1	1.1	0.09	0.99
10	1;2’	0.9	1.1	1.0	1.3	1.2	1.1	0.17	1.56
11	1;6	0.5	0.8	0.8	1.0	0.7	0.8	0.24	0.14
12	2;5	0.8	0.7	0.8	1.1	0.8	0.9	0.17	0.31
13	2;5’	0.7	0.8	0.8	1.1	0.9	0.9	0.16	0.65
14	2;4	0.8	0.7	0.8	0.8	0.9	0.8	0.06	0.89
15	2;6	0.7	0.3	0.9	0.9	1.0	0.8	0.35	0.75

**Table 4 sensors-23-01114-t004:** Modal shape deviation.

Mode		Modal Shape Variation							Unc-ty
Nr	Type	Sp 1	Sp 2	Sp 3	Sp 4	Sp 5	Typ.Sp	STD (1...5)	
1	2	4	5	6	7	8	9	10	11
1	1;1	1.0%	1.5%	1.0%	1.3%	1.4%	1.2%	0.25%	0.4%
2	1;1’	0.6%	2.1%	0.5%	0.5%	0.9%	0.9%	0.66%	0.2%
3	1;4	0.3%	0.6%	0.7%	1.1%	0.8%	0.7%	0.29%	0.6%
4	1;4’	0.4%	0.3%	1.6%	0.2%	0.3%	0.6%	0.57%	0.7%
5	1;3	0.5%	1.3%	0.4%	0.3%	1.2%	0.8%	0.47%	0.5%
6	1;3’	0.6%	0.8%	0.8%	1.3%	1.8%	1.1%	0.49%	0.5%
7	1;5	0.5%	0.3%	0.6%	0.4%	0.6%	0.5%	0.14%	0.7%
8	1;5’	0.3%	0.6%	0.4%	0.3%	0.5%	0.4%	0.14%	0.8%
9	1;2	0.6%	0.8%	0.9%	1.2%	1.2%	0.9%	0.23%	0.3%
10	1;2’	0.8%	0.6%	1.2%	0.8%	0.9%	0.9%	0.21%	0.3%
11	1;6	0.4%	0.7%	0.4%	1.0%	0.3%	0.6%	0.30%	0.5%
12	2;5	0.5%	0.5%	0.4%	0.8%	0.6%	0.6%	0.15%	0.6%
13	2;5’	0.3%	0.3%	0.3%	0.9%	1.0%	0.6%	0.35%	0.8%
14	2;4	0.5%	1.0%	0.7%	0.4%	0.4%	0.6%	0.26%	1.2%
15	2;6	0.7%	0.3%	0.3%	0.3%	0.3%	0.4%	0.19%	2.3%

## Data Availability

Not applicable.

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
