# Peer review of "Health Monitoring of Serial Structures Applying Piezoelectric Film Sensors and Modal Passport"

_sensors, 2023, doi:10.3390/s23031114_

Round 1

Reviewer 1 Report

This paper deals with Operational Modal Analysis (OMA) of 5 composite structures, in order to study the influences on Structural Health Monitoring (SHM) of serial samples and consider the similarities and differences between samples. 

As it is mentioned in the introduction, the work will be focused on three tasks:

- the optimum sensor network solution for serial products
- the methodology for considering similarity and difference of modal properties between serial structures
- practical verification of technical and methodical solutions.

The first one has not been addressed in this paper. The other two issues have been experimentally studied, but it is a lack of methodology and parameters under study.

Next, I will explain some of the deficiencies that needs to be improved or clarified:

- From the point 2 ("Materials and methods"), the information about the types of sensors used in OMA is wrong and incomplete. For Example, Strain Gauges do not usually embed in composites, and the cabling always has a significant influence in the aerodynamic profile. I recommend avoiding this issue as it has not enough relevance in the topic, or include a reference with proved information about the sensors.

- Figure 2: Box 2 (Signal data recording procedures) is not connected, and sensing methods and errors should be taken into account in the methodology, as experimental errors should be also taken into account.

- Page 5: The influence of ambient is crucial (not only temperature, also stress (not wind), humidity), and has not been considered. It is not clear how it is considered. Also, it is unclear how model enhancement it is performed (in groups or separately).

- FEM model requires further information about boundary conditions. Some properties, such young modulus or density should be detailed, and units should be properly presented.

- From point 3 ("Experiments"), it is a problem related with the structure under study: the symmetry. Symmetry results in coupled modes and makes difficult the OMA analysis. As a results, the modes presented in Table 1 are mostly coupled modes. For example, 3 and 4, 5 and 6, 7 and 8, 9 and 10, 11 and 12, 13 and 14, 15 and 16 and 17 and 18 are coupled modes. Even after this coupling is mentioned, not in all the previous modes.

- From point 3.3.1 the piezoelectric film sensor type should be included.

- From Point 3.4 (model estimation of specimens), the differences between experimental estimation and FEM model are, in some cases, huge. In my opinion the model do not represent the experiment, probably due to the boundary conditions.

For the previous reasons, I do not consider that the discussion and the conclusions are substantiated. For example, the conclusions mention the cost effective and reliability of piezofilm sensors for OMA, even if it is not mentioned the huge efforts for instrumentation the sensors on the specimens. It is also mentioned the SHM applications of OMA, but it is not analysed the capability of damage detection and the improvements of the proposed analysis.

In general, I consider a hard work under this paper, but it is a lack of analysis to support any useful conclusion on the topic. FEM Model and analysis should be improved, and the already performed work should be the basis for the study of the influence of similarities and differences and how to manage with them to improve OMA for SHM.

Reviewer 2 Report

This article introduces the concept of modal passport applied to piezo electric sensors in details.

 The Operational Modal Analysis (OMA) techniques is introduced in this paper, and the practical methods are illustrated with the sensing application of aerospace structure. There are several questions to be clarified:

1. Where is the innovation of the paper? What practical problem, which cannot be solved by nomal methods, has been solved.

2. The actual test result datas and test conditions are very complex. How to deal with this complexity

Round 2

Reviewer 1 Report

Thank you for clarifying most of the concerns mentioned in the previous review. However, the main concern is still no analysed: the damage detection capability, providing information about the minimum damage(s) detection size and severity (hole, delamination) and the influence in the different modes. Damage should be theoretically and/or experimentally analysed to support the conclusions and analyse the sensitivity.

Also, I do not agree with the "embedded" definition, as in the bibliography is always referred as inside the material (e.g. between composite layers)

Reviewer 2 Report

The authors have addressed all my previous comments and substantially improved the paper. I recommend to publish the manuscript as it is.

Round 3

Reviewer 1 Report

Thank you for your clarifications about the full frame of the project and the previous results. In my opinion it will be useful to present all toguether to understand the full work performed and the results.

However, I am strongly disagree about the use of "embedded". It is not a "point of view", and it is the different between inside the material (inside or between layers) or just fix out of the material. This must be corrected in the final version.
